# Liposomal Formulations of a New Zinc(II) Complex Exhibiting High Therapeutic Potential in a Murine Colon Cancer Model

**DOI:** 10.3390/ijms23126728

**Published:** 2022-06-16

**Authors:** Nádia Ribeiro, Melissa Albino, Andreia Ferreira, Cristina Escrevente, Duarte C. Barral, João Costa Pessoa, Catarina Pinto Reis, Maria Manuela Gaspar, Isabel Correia

**Affiliations:** 1Centro Química Estrutural, Departamento de Engenharia Química, Instituto Superior Técnico, Universidade de Lisboa, 1049-001 Lisboa, Portugal; nadia.ribeiro@tecnico.ulisboa.pt (N.R.); joao.pessoa@tecnico.ulisboa.pt (J.C.P.); 2Research Institute for Medicines (iMed.Ulisboa), Faculty of Pharmacy, Universidade de Lisboa, 1649-003 Lisboa, Portugal; melissateixeira@campus.ul.pt (M.A.); catarinareis@ff.ulisboa.pt (C.P.R.); 3iNOVA4Health, NOVA Medical School (NMS), Faculdade de Ciências Médicas (FCM), Universidade Nova de Lisboa, 1169-056 Lisboa, Portugal; andreia.ferreira@nms.unl.pt (A.F.); cristina.escrevente@nms.unl.pt (C.E.); duarte.barral@nms.unl.pt (D.C.B.); 4IBEB, Faculdade de Ciências, Universidade de Lisboa, Campo Grande, 1749-016 Lisboa, Portugal

**Keywords:** zinc complexes, hydrazone, liposomes, colorectal cancer, 2D and 3D cell models, in vivo studies

## Abstract

Colorectal cancer is the second leading cause of cancer-related mortality. Many current therapies rely on chemotherapeutic agents with poor specificity for tumor cells. The clinical success of cisplatin has prompted the research and design of a huge number of metal-based complexes as potential chemotherapeutic agents. In this study, two zinc(II) complexes, [ZnL_2_] and [ZnL(AcO)], where AcO is acetate and L is an organic compound combining 8-hydroxyquinoline and a benzothiazole moiety, were developed and characterized. Analytical and spectroscopic studies, namely, NMR, FTIR, and UV-Vis allowed us to establish the complexes’ structures, demonstrating the ligand-binding versatility: tetradentate in [ZnL(AcO)] and bidentate in [ZnL_2_]. Complexes were screened in vitro using murine and human colon cancer cells cultured in 2D and 3D settings. In 2D cells, the IC_50_ values were <22 µM, while in 3D settings, much higher concentrations were required. [ZnL(AcO)] displayed more suitable antiproliferative properties than [ZnL_2_] and was chosen for further studies. Moreover, based on the weak selectivity of the zinc-based complex towards cancer cell lines in comparison to the non-tumorigenic cell line, its incorporation in long-blood-circulating liposomes was performed, aiming to improve its targetability. The resultant optimized liposomal nanoformulation presented an I.E. of 76% with a mean size under 130 nm and a neutral surface charge and released the metal complex in a pH-dependent manner. The antiproliferative properties of [ZnL(AcO)] were maintained after liposomal incorporation. Preliminary safety assays were carried out through hemolytic activity that never surpassed 2% for the free and liposomal forms of [ZnL(AcO)]. Finally, in a syngeneic murine colon cancer mouse model, while free [ZnL(AcO)] was not able to impair tumor progression, the respective liposomal nanoformulation was able to reduce the relative tumor volume in the same manner as the positive control 5-fluorouracil but, most importantly, using a dosage that was 3-fold lower. Overall, our results show that liposomes were able to solve the solubility issues of the new metal-based complex and target it to tumor sites.

## 1. Introduction

Recent statistics compiled by GLOBOCAN 2020 indicate that colorectal cancer (CRC) ranks third in incidence and second in cancer-related mortality [1]. CRC, also known as colorectal adenocarcinoma, derives from the glandular epithelial cells of the intestine that have undergone malignant transformation [2]. Chemotherapy remains one of the most important therapeutic options in the management of CRC. 5-Fluorouracil (5-FU), trifluridine, irinotecan, and oxaliplatin are drugs used in chemotherapeutic regimens for CRC treatment [3]. 5-FU and trifluridine are pyrimidines, irinotecan contains a quinoline in its structure, and oxaliplatin is a metallodrug. However, the efficacy of these drugs is often hampered by drawbacks associated to its use, such as low specificity and inadequate pharmacokinetic and biodistribution profiles, as recently reviewed [4]. Thus, increasing incidence combined with the limited effectiveness of current therapeutic strategies demands the development of novel therapeutic agents with high potency and distinct mechanisms of action.

Ligands in coordination compounds, such as cisplatin and other metallodrugs, play crucial roles: tuning the complexes’ redox properties, modulating the membrane permeability, and hence directing the complex to cellular compartments and possibly exhibiting intrinsic cytotoxicity when dissociated from the metal ion. Alternatively, the metal ion can stabilize the ligand and act simply as a carrier to the target. Ideally, the effects of both entities in a metal complex may synergistically result in enhanced selectivity and activity, possibly by inducing new effective mechanisms of action.

It is a common practice in academia and industry to combine different pharmacological active compounds in the development of new drugs [5]. 8-Hydroxyquinoline (8HQ) presents an excellent ability to bind diverse metal ions and has been used in their gravimetric analysis and separation [6]. Additionally, it is considered a privileged structure for drug development due to its numerous reported biological activities [7,8,9,10]. Our hypothesis is that extending the coordination ability of 8HQ by attaching additional donor groups adjacent to the quinoline nitrogen will modify the properties of the metal-based complexes, increasing their stability and activity by the assembly of different bioactive organic moieties. It has been reported that the introduction of substituents in this carbon atom can lead to compounds with interesting and rich coordination chemistry [11,12,13]. Condensation reactions yielding Schiff bases [14] are effective and synthetically accessible strategies to perform the task, driving the design of new chemotherapeutic agents [15].

Benzothiazole is another emerging privileged structure in anticancer research [16,17,18]. The synthesis and anticancer activity of 2-benzothiazole hydrazones towards human cancer cell lines has been reported [19], with most compounds showing a good cytotoxic profile and a few revealing IC_50_ values in the nM range in HL-60 (leukemia cancer cells). Later, another 2-hydrazinobenzothiazole derivative showed anticancer potential against melanoma cells due to DNA fragmentation induction [20]. Tayler and co-workers evaluated the antiproliferative properties and artificial nuclease activities of a set of Cu(II)-8HQ-benzothiazole complexes [21]. Besides copper, the zinc benzothiazole Schiff bases also presented very promising results [22]. In the present work, a new organic ligand precursor was synthesized, conjugating the two biologically active modules, 8HQ and benzothiazole. Zinc was chosen as the metal core due to the therapeutic potential of Zn complexes and particularly its Schiff bases, [23] and thus, two novel metal complexes were synthesized and are reported. In fact, it is not the first time that these two types of structures were conjugated for anticancer therapy. A benzothiazolyl quinoline fluorescent scaffold was developed as a cancer theranostic with good quantum yield in water and significant cellular uptake, potency, and selectivity for cancer cell lines [24].

Despite the promising results concerning the cytotoxic potential of metal-based complexes, their translation into clinical trials is frequently hampered by non-selective toxicity and a lack of stability and solubility in aqueous media [4]. The use of drug delivery systems can overcome these drawbacks, increasing the therapeutic efficacy of the incorporated metal-based compound by promoting a preferential delivery [25,26]. Lipid-based nanocarriers have shown great potential in delivering antitumor agents following adsorption onto their surfaces or incorporation within their cores. In fact, nanocarriers, depending on their physicochemical properties, stabilize loaded compounds, improving the targetability of the delivery system and concomitantly increase the concentration of the incorporated drug at affected sites. Moreover, this particular in vivo fate avoids drug accumulation in non-affected sites and consequently reduces adverse side effects [27]. Liposomes are the most well-known and versatile lipid-based nanocarriers due to their unique characteristics. They can incorporate both hydrophilic and hydrophobic compounds in aqueous and lipid bilayer compartments, respectively. They present high biocompatibility, biodegradability, low immunogenicity, and the ability to protect the entrapped drug from premature degradation, enhancing its stability as well as its solubility [4]. Additionally, many liposomal formulations (more than 50) have already been approved for clinical use against various diseases, namely, cancer [25,28]. In fact, liposomal formulations of cisplatin and its derivatives have been developed for the treatment of multiple malignancies [29], including CRC [30]. The list of anticancer drugs incorporated in liposomes for the treatment of CRC is increasing and includes 5-FU and irinotecan, demonstrating the potential of nanoliposomes in CRC chemotherapy [25].

It has been thoroughly described that nanoliposomes can accumulate in regions of enhanced vascular permeability, such as those found in inflammatory and tumor sites, as a result of the enhanced permeation and retention (EPR) effect [31]. To take advantage of this effect, long bloodstream circulating times are required. This can be achieved through the addition of the polymer polyethylene glycol (PEG) in the lipid composition [32]. Furthermore, the microenvironment of tumors is known to be slightly acidic, exhibiting pH values around 6, in contrast to the healthy pH of 7.4 [33]. Liposomes can be designed to have a pH-triggered compound release. A commonly used strategy to develop pH-sensitive liposomes is the inclusion of the phospholipid dioleoyl phosphatidyl ethanolamine (DOPE) in combination with a lipid such as cholesteryl hemisuccinate (CHEMS). DOPE, due to its unsaturated chains, forms an inverted hexagonal phase instead of bilayers that are stable at a neutral pH. However, in a slightly acidic microenvironment, CHEMS becomes protonated, leading to membrane destabilization and promoting the release of the loaded compounds [34,35]. Thus, the combination of these two features allows a pH-triggered zinc compound release from long-blood-circulating nanoliposomes. This strategy was used in the present system after the selection of the best compound to incorporate in the nanoliposomes. The antiproliferative properties of metal-based complexes were evaluated following their incubation with colon cancer cells in 2D and 3D settings. Although 2D monolayer cells are most frequently used to screen compound cytotoxicity, 3D settings allow a more accurate representation of the in vivo tumor characteristics. Hence, they are more apt and reliable to select the most promising compounds and/or formulations. Finally, a syngeneic colon cancer murine model was also developed to test the in vivo efficacy of the free and liposomal formulations of the selected zinc complex.

## 2. Experimental Section

### 2.1. Materials and Apparatus

2-Carbaldehyde-8-hydroxyquinoline, 2-hydrazinobenzothiazole, and Zn(II) acetate were obtained from Sigma and used without further purification. Phosphate-buffered saline (PBS) was obtained from Sigma as readily soluble tablets, and HEPES (4-(2-hydroxy ethyl)-1-piperazine ethane sulfonic acid) was purchased from Sigma. The water used in all studies with biomolecules was double-deionized in a Milli-Q system. The pure phospholipids dioleoyl phosphatidyl choline (DOPC), dimyristoyl phosphatidyl choline (DMPC), and DOPE as well as cholesteryl hemisuccinate (CHEMS) and PEG 2000 covalently linked to distearoyl phosphatidyl ethanolamine (DSPE-PEG) were purchased from Avanti Polar Lipids (Alabaster, AL, USA). All other chemical products and solvents used were of analytical grade. Cell culture reagents and kits were purchased from Thermo Fisher Scientific, Waltham, MA, USA. Electronic absorption spectra (UV-Vis) were recorded with a Perkin Elmer Lambda 35 spectrophotometer. Elemental analysis for C, H, N, and S, were carried out on a FISONS EA 1108 CHNS-O apparatus at Laboratório de Análises of Instituto Superior Técnico. ^1^H and ^13^C NMR spectra were recorded at ambient temperature on a Bruker Avance II + 400 (UltraShieldTM Magnet) spectrometer operating at 400.13 MHz for protons and at 81 MHz for carbon. The chemical shifts are reported in ppm using tetramethylsilane as an internal reference. Infrared spectra were recorded on a JASCO FT/IR 4100 spectrometer (JASCO International Co., Ltd., Hachioji, Tokyo, Japan), and a 500-MS LCQ Fleet mass spectrometer with an ESI source (Thermo Scientific, Whaltham, MA, USA) interfaced with an HPLC-DAD (Varian, Agilent Technologies, Inc., Santa Clara, CA, USA) was used to measure the ESI-MS spectra of the methanolic solutions of the compounds in both the positive and negative ion modes.

### 2.2. Methods

#### 2.2.1. Synthesis of HL

In a reaction vessel, 1 mol equivalent (1 eq.) of 2-carbaldehyde-8-hydroxyquinoline was solubilized in ethanol (EtOH), and some drops of glacial acetic acid were added. Then, 1 eq. of 2-hydrazinobenzothiazole was added with constant stirring. The mixture was left at room temperature for 6 h; a yellow solid formed, was collected by filtration, and was washed with ice-cold EtOH. The solid was dried under vacuum in a desiccator over silica gel. Yield: 93%. Experimental elemental analysis: C, 61.6%; H, 3.8%; N, 16.8%; S, 9.6% (calcd. for C_17_H_12_N_4_OS∙0.5H_2_O: C, 61.99%; H, 3.98%; N, 17.01%; S, 9.73%). ESI-MS (*m*/*z*) 321.20 [C_17_H_12_N_4_OS + H]^+^; 319.47 [C_17_H_12_N_4_OS − H]^−^. ^1^H NMR (400 MHz, DMSO_*d*_6_, δ (ppm)): 12.75 (NH); 9.82 (OH); 8.34 (HC=N); hydroxyquinoline aromatic moiety: 7.13, 7.44, 7.41, 8.35, 8.04; benzothiazole moiety: 7.83, 7.16, 7.34, 7.5. ^13^C NMR (400 MHz, DMSO_*d*_6_, δ (ppm)): 144.10 (imine); hydroxyquinoline moiety: 138.13, 153.57, 112.16, 128.08, 117.80, 128.58, 136.6, 117.25, 151.42; benzothiazole moiety: 161.86, 130.19, 121.65, 122.08, 126.11, 120, 151.4. Soluble in dimethylformamide (DMF) and dimethylsulfoxide (DMSO).

#### 2.2.2. Synthesis of the Zinc(II) Complexes

[ZnL(AcO)]—In a two-entry reaction vessel, 1 eq. of [Zn(CH_3_COO)_2_·2H_2_O] was dissolved in EtOH and kept warm. In a glass vial, 1 eq. of HL was suspended in EtOH, deprotonated with methanolic KOH, added dropwise to the zinc solution, and the mixture was allowed to reflux for 5 h. After cooling and refrigeration, the solid was collected by filtration, washed with ice-cold EtOH, and dried under vacuum in a desiccator over silica gel. Yield: 78.4 mg (70.7%), red solid. Experimental elemental analysis: C, 51.0%; H, 2.8%; N, 12.5%; S, 7.6% (calcd. for C_19_H_14_N_4_O_3_SZn: C, 51.42%; H, 3.18%; N, 12.62%; S, 7.22%). ESI-MS (*m*/*z*) 460.69 [ZnL(AcO)]+NH_4_]^+^; 455.29 [ZnL+2Cl]^−^. ^1^H NMR (400 MHz, DMSO_*d*_6_, δ (ppm)): 11.96 (1H, NH); 9.90 (1H, HC=N); 8.37 (1H); 8.20 (1H); 8.03 (1H); 7.53 (1H); 7.39 (1H); 7.21 (1H); 6.99-6.95 (3H); 1.87 (3H, AcO). ^13^C NMR (400 MHz, DMSO_*d*_6_, δ (ppm)): 144.9 (HC=N); 162.07; 151.8; 148.5; 139.02; 138.3; 129.3; 126.6; 125.4; 121.06; 120.1; 117.35; 115.42; 113.13; 110.37; 21.32 (C, AcO). Soluble in DMF and DMSO.

[ZnL_2_]—In a two-entry vessel, 2 eq. of HL were suspended in EtOH and deprotonated with KOH in MeOH. In a glass vial, 1 eq. of the zinc acetate was dissolved in EtOH and added dropwise to the ligand solution. The dark orange mixture was left to reflux for 5 h. The mixture was left overnight in the freezer. The light orange solid was collected by filtration, washed with ice-cold EtOH, and dried under vacuum in a desiccator over silica gel. Yield: 108.8 mg (77.3%), light orange solid. Experimental elemental analysis: C, 56.5%; H, 3.2%; N, 14.8%; S, 9.0% (calcd. for C_34_H_22_N_8_O_2_S_2_Zn∙1H_2_O∙0.25EtOH: C, 56.48%; H, 3.50%; N, 15.27%; S, 8.74%). ESI-MS (*m*/*z*) 705.8 [ZnL_2_+H]^+^; 703.7 [ZnL_2_−H]^−^; 739.1 [ZnL_2_+Cl]^−^. ^1^H NMR (400 MHz, DMSO_*d*_6_, δ (ppm)): 8.34 (2H, HC=N); 9.79 (2H, OH); 8.35 (2H); 8.05 (2H); 7.83 (2H); 7.45-7.41 (6H); 7.34 (2H); 7.17 (2H); 7.12 (2H). ^13^C NMR (400 MHz, DMSO_*d*_6_, δ (ppm)): 143.55 (HC=N); 153.2; 151.7; 138; 136.41; 130.08; 128.5; 128.0; 125.98; 121.96; 121.51; 117.55; 116.99; 111.95. Soluble in DMF and DMSO.

#### 2.2.3. Stability Assays under Aqueous Conditions 

Stock solutions of each complex were prepared in DMSO and diluted in the aqueous PBS (0.01 M in phosphate, 0.138 M NaCl, 0.0027 M KCl, and pH 7.4 at 25 °C), ensuring that the organic solvent content was less than 1% (*v*/*v*). For the assays with equimolar amounts of bovine serum albumin (BSA), stock solutions of 20 mg of lyophilized protein were prepared in 5 mL of aqueous buffer and allowed to hydrate for 24 h in the refrigerator. The samples were monitored by UV-Vis absorption spectroscopy for 6 consecutive hours, and a final measurement was taken 24 h after mixing.

#### 2.2.4. Interaction Studies with Biomolecules

##### Fluorescence Assays with Bovine Serum Albumin

The fluorescence emission spectra of a solution of ca. 1.5 μM BSA (in PBS) were recorded between 310 nm and 575 nm with λ_ex_ = 295 nm upon successive additions of the complexes directly to the cuvette (optical path of 1.0 cm). The UV-Vis spectra of each sample were measured and used to correct the emission spectra, minimizing inner filter and reabsorption phenomena interference [36,37]. Blank assays (with no BSA added) for each sample, consisting of solutions with the same concentration of complex, were recorded and subtracted from the corresponding emission spectra containing the protein. 

##### Circular Dichroism Assays with Bovine Serum Albumin

The concentration of BSA in the experiments was ca. 20–25 μM (in PBS). The presented circular dichroism (CD) spectra are an average of three scan accumulations, recorded between 300 and 500 nm, with a bandwidth of 1 nm and a response of 2 s using a scanning speed of 100 nm/min. Successive additions of the complexes’ stock solutions in DMSO, performed directly to the cuvette (1.0 cm), allowed the acquisition of the CD spectra under the same conditions after 10 min of equilibration. The induced CD spectra were obtained as the CD spectra of compound-BSA mixtures after the subtraction of the CD spectrum of BSA alone.

##### Fluorescence Competition Assays with Ethidium Bromide and DNA

Prior to titration with the complexes, *calf thymu*s DNA (*ct*DNA) was saturated with ethidium bromide (EB). Selected accurate volumes of the *ct*DNA and EB stock solutions were added in the fluorescence cuvette, with a total volume of 2500 μL and completed with HEPES buffer (0.1 M HEPES, pH 7.4, 0.15 M KCl), to ensure an EB/DNA ratio of 0.8. The fluorescence emission spectrum was recorded between 520 nm and 750 nm with λ_ex_ = 510 nm. Successive aliquots of the complexes’ stock solutions were added directly to the cuvette, and fluorescence emission spectra were recorded for each of them. Blank assays were performed for each complex in which the fluorescence under the same concentrations for *ct*DNA and the complex were recorded and subtracted from each corresponding emission spectrum.

#### 2.2.5. Cell Culture Conditions

Culture media and antibiotics were obtained from Invitrogen (Thermo Fisher Scientific, Waltham, MA, USA). Murine colon cancer CT-26 (ATCC^®^ CRL-263TM) cells were cultured in RPMI-1640 supplemented with 10% fetal bovine serum (FBS), 100 IU/mL of penicillin, and 100 μg/mL streptomycin (Gibco, Thermo Fisher Scientific, Waltham, MA, USA), hereafter designated as complete medium. Human colon cancer HCT-116 cells (ATCC^®^ CCL-247TM) and the human immortalized keratinocyte cell line HaCaT (CLS Cell Line Service GmbH, Eppelheim, Germany) were cultured in Dulbecco’s Modified Eagle’s medium (DMEM) with high-glucose (4500 mg/L) GlutaMax (Sigma-Aldrich, St. Louis, MO, USA) supplemented with 10% FBS, 100 IU/mL penicillin, and 100 μg/mL streptomycin (Gibco, Thermo Fisher Scientific, Waltham, MA, USA), hereafter designated as complete medium. The cell lines were kept at 37 °C with a 5% CO_2_ atmosphere. The maintenance of cell cultures was performed every 2–3 days until the cells reached a confluency of about 80%.

##### Spheroids

For the formation of HCT-116 and CT-26 spheroids, 96-well round bottom plate (Corning) wells were coated with 50 µL of 1.5% agarose and allowed to solidify for 20 min. Next, 200 µL containing 3000 cells per well were seeded and incubated at 37 °C for 72 h to form spheroids. In the case of HCT-116 spheroids, cells were seeded in DMEM complete medium. Moreover, 100 µL of the medium was replaced 24 h after seeding. Concerning the CT-26 spheroids, cells were seeded in 200 µL of RPMI complete medium. Again, 100 µL of the medium was replaced 24 h after seeding.

#### 2.2.6. Antiproliferative Activity

##### 2D Setting

The cell viability was evaluated by the 3-(4,5-dimethylthiazol-2-yl)-2,5-diphenyl-2H-tetrazolium bromide (MTT) assay, as it assesses metabolic activity that only occurs in viable cells in the presence or absence (negative control) of increasing concentrations of two zinc complexes, [ZnL(AcO)] and [ZnL_2_], and of [ZnL(AcO)] in the free or liposomal forms. As a positive control group, the antiproliferative properties of 5-FU were also evaluated. After reaching confluency, cells were gently trypsinized and counted using a hemocytometer and by the trypan-blue exclusion of dead cells. Then, a cell suspension at a density of 5 × 10^4^ cells/mL (200 μL) was seeded in 96-well culture plates and left in culture for 24 h. Afterwards, the supernatant was removed, and cells were incubated with the formulations under study at concentrations ranging from 5 to 40 μM for 48 h. Then, the medium was removed, and cells were washed twice with 200 μL of phosphate-buffered saline (PBS). Subsequently, 50 μL of the MTT solution (0.5 mg/mL), prepared in the appropriate serum-free medium, was added to each well, followed by a 3 h incubation at 37 °C and 5% CO_2_. Finally, formazan crystals were solubilized in DMSO, and the absorbance (Abs) was measured at 570 nm using a Model 680 microplate reader (Bio-Rad, Hercules, CA, USA).

##### 3D Setting

First, the cellular cytotoxicity on CT-26 and HCT-116 spheroids was evaluated in the presence of increasing concentrations of the zinc complexes [ZnL(AcO)] and [ZnL_2_] using the lactate dehydrogenase (LDH) cytotoxicity detection kit (Takara, Kusatsu, Japan), as the LDH present in the medium was proportional to cell lysis. The negative controls consisted of spheroids in complete medium and in complete medium containing 1.6% DMSO, which corresponded to the maximum concentration of DMSO present in the metal-based complex solutions tested. For [ZnL(AcO)] and [ZnL_2_], a range of concentrations from 5 to 20 µM was added to the spheroids 72 h after formation. Following an incubation period of 72 h, 30 µL of medium was removed from each well and kept in Eppendorf tubes. These samples contained the released LDH caused by the incubation period, thus corresponding to the induced cell death (A). Then, 22 µL of medium and 8 µL of 25% Triton X-100 (p/v) were added to each well in order to dissociate the spheroids into single cells and facilitate cell lysis, enabling LDH release into the medium. These samples were also kept in Eppendorf tubes (B). Afterwards, all samples were kept at 4 °C for at least 1 h and vortexed and centrifuged for 3 min at 10× *g*. Finally, 100 µL of Milli-Q water was plated into a new 96-well plate as well as 10 µL of each sample and 100 µL of a mixture of solutions 1 and 2, included in the kit. Then, the Abs was measured at 490 nm using a SpectraMax i3x microplate reader (Molecular Devices, San Jose, CA, USA). Then, the absorbance of all samples was also measured at the reference wavelength of 655 nm. The cytotoxicity was calculated according to the following equation:(1)$$\mathrm{Cytotoxicity}\;\left(\%\right)=\frac{\mathrm{Induced}\;\mathrm{cell}\;\mathrm{death}\;\left(A\right)}{\mathrm{Total}\;\mathrm{Cell}\;\mathrm{death}\;\left(A+B\right)}\times 100$$

The CellTiter-Glo^®^ 3D kit (Promega, Madison, WI, USA) was also used to assess the viability of the HCT-116 and CT-26 spheroids in the presence or absence (negative control) of increasing concentrations of the free and liposomal forms of the zinc complex [ZnL(AcO)]. This kit allows the detection of adenosine triphosphate, which is proportional to the number of viable cells thus allowing the estimation of cell proliferation. First, solutions of [ZnL(AcO)] in the free and liposomal forms ranging from 20 to 200 µM were added to the spheroids after 72 h of spheroid formation. After an incubation period of 120 h, the spheroids were transferred into a 96-well opaque-walled plate. Then, the CellTiter-Glo^®^ 3D reagent was added into each well at the same volume of cell culture medium. Afterwards, the contents were mixed vigorously for 5 min to induce cell lysis, and the plate was incubated at room temperature for 25 min to stabilize the luminescent signal. Finally, the luminescence was measured using a SpectraMax i3x microplate reader (Molecular Devices, San Jose, CA, USA).

#### 2.2.7. Cell Death Analysis

The Alexa Fluor^®^-488 Annexin V/Dead Cell Apoptosis Kit (Thermo Fisher Scientific, Waltham, MA, USA) was used to determine cell death induced by the zinc complex [ZnL(AcO)]. Briefly, cells were seeded at a density of 1 × 10^5^ cells/mL in 6-well plates and left in culture at 37 °C and 5% CO_2_ for 24 h. Afterwards, the supernatant was removed, and cells were incubated either with [ZnL(AcO)] at a concentration of 20 μM or complete medium for 24 h. Then, cells were harvested and washed in cold PBS. The cell density was determined with a hemocytometer, and cells were diluted in 1× annexin-binding buffer to 1 × 10^6^ cells/mL for each condition. Moreover, the cell suspension was incubated with 5 µL of Alexa Fluor-488 Annexin V and 1 µL of 100 µg/mL propidium iodide (P.I.) for 15 min at room temperature. After the incubation period, 400 µL of 1× annexin-binding buffer was added. Data acquisition was performed in a FACS CANTO II (BectonDickinson, Franklin Lakes, NJ, USA) flow cytometer, and at least 30,000 cells were analyzed using FlowJo software (version 10.1r7).

#### 2.2.8. Cell Cycle Analysis 

Flow cytometry was used to assess the potential effects of [ZnL(AcO)] on cell cycle progression, according to a published procedure [35]. Briefly, cells were seeded at a density of 1.5 × 10^5^ cells/mL in 6-well plates and left in culture at 37 °C and 5% CO_2_ for 24 h. Afterwards, the supernatant was removed, and the cells were incubated with either [ZnL(AcO)] at a concentration of 5 or 10 μM or complete medium for 24 h. The cells were then washed in PBS and collected by centrifugation at 200× *g* for 5 min at room temperature. Moreover, the cells were suspended in ice-cold PBS and fixed with an equal volume of ice-cold 70% ethanol (−20 °C), added drop wise under gentle vortexing. The samples were stored at 4 °C until data acquisition. Before acquisition, cells were centrifuged at 800× *g* for 5 min and washed twice with PBS. Finally, 50 μL of RNAse (Sigma-Aldrich) at 100 μg/mL and 200 μL of P.I. (Sigma-Aldrich) at 50 µg/mL were added just before analysis. Data acquisition was performed using the FACS Canto II (BectonDickinson, Franklin Lakes, NJ, USA) flow cytometer, and at least 30,000 cells were acquired per condition. Data analysis was carried out using FlowJo software (version 10.1r7).

#### 2.2.9. Liposomal Formulations

##### Preparation of Liposomal Formulations

Liposomes composed of the selected phospholipids were prepared by the dehydration–rehydration method, followed by an extrusion step to reduce and homogenize liposomal formulations [34,35,38]. The constituents for liposome preparation consisted of a lipid mixture of DOPC, DOPE, and CHEMS to confer pH-sensitive properties and DSPE-PEG to achieve long blood-circulation times using different molar ratios. For all the nanoformulations, an initial lipid concentration of 30 μmol/mL was used. Briefly, the selected lipid mixture and the zinc complex [ZnL(AcO)] (at 0.5 or 1 mg/mL) were dissolved in chloroform, and the solvent evaporated in a Buchi R-200 rotary evaporator (Flawil, Switzerland) to obtain a thin lipid film in a round-bottom flask. To prepare rhodamine (Rh)-labelled liposomes, rhodamine covalently linked to phosphatidyl ethanolamine was included in the lipid mixture at 0.1 mol% related to the total lipids before the evaporation step. The obtained lipid film was hydrated with water, and the formed suspension was frozen (−70 °C) and lyophilized (freeze-dryer, Edwards, CO, USA) overnight. The rehydration of the lyophilized powder was performed in HEPES buffer at pH 7.4 (10 mM HEPES and 140 mM NaCl) in two steps to enhance the Zn-complex incorporation [35,38]. First, a volume of up to two tenths of the initial dispersion volume was added. After 30 min, pH 7.4 HEPES buffer was added up to the original volume. The rehydration process was performed above the phase transition temperature (T_c_) of the main phospholipids. The formed liposomal suspensions were then filtered under nitrogen pressure (10–500 lb/in) through polycarbonate membranes of an appropriate pore size until the achievement of an average vesicle size below 150 nm using an extruder device (Lipex: Biomembranes Inc., Vancouver, BC, Canada). The separation of the non-incorporated metal complex was performed by gel filtration (Econo-Pac^®^ 10DG; Bio-Rad Laboratories, Hercules, CA, USA), followed by ultracentrifugation at 250,000× *g* for 120 min at 15 °C in a Beckman LM-80 ultracentrifuge (Beckman Instruments, Inc, Fullerton, CA, USA). Finally, the pellet was suspended in pH 7.4 HEPES buffer.

##### Physicochemical Characterization of Liposomal Formulations

Liposomes were characterized in terms of the mean size, polydispersity index (PdI), surface charge, and incorporation parameters. Concerning the surface charge, mean size, and PdI, the formulations were diluted in pH 7.4 HEPES (1:100, *v*/*v*) and a Zetasizer Nano ZS (Malvern Instruments, Malvern, UK) was used to perform the measurements by dynamic light scattering. The loading capacity was defined as the final [ZnL(AcO)]-to-lipid ratio ([ZnL(AcO)]/Lip)_f_, and the incorporation efficiency (I.E.) in percentage was determined according to the following equation:(2)$$I.E.\;\left(\%\right)=\frac{\left(\frac{\left[\mathrm{Zn}\left(L\right)\left(\mathrm{AcO}\right)\right]}{\mathrm{Lip}}\right)f}{\left(\frac{\left[\mathrm{Zn}\left(L\right)\left(\mathrm{AcO}\right)\right]}{\mathrm{Lip}}\right)i}\times 100$$

[ZnL(AcO)] was quantified spectrophotometrically at 393 nm after the disruption of the nanoliposomes with ethanol [34,35,38]. The linearity of the calibration curves was ensured using [ZnL(AcO)] standards ranging from 5 to 30 μg/mL. The phospholipid content was determined using the method described by Rouser [39].

##### Assessment of pH-Sensitive Properties

Freshly prepared liposomes of DOPC:CHEMS:DOPE:DSPE-PEG loading the zinc complex [ZnL(AcO)] were incubated in HEPES buffer at three different pH values, 4.5, 6, and 7.4, for 90 min at 37 °C under stirring [34,35]. At the end of the incubation period, the separation of the released metal-based complex from the liposomes was performed by gel filtration in an Econo-Pac^®^ 10DG (Bio-Rad Laboratories, Hercules, CA, USA), followed by ultracentrifugation at 250,000× *g* for 120 min at 15 °C in a Beckman LM-80 ultracentrifuge. The pellets were suspended in pH 7.4 HEPES buffer according to the initial volume, and the [ZnL(AcO)] and phospholipid contents were determined spectrophotometrically, as described above. The stability was defined as the ratio in percentage between the [ZnL(AcO)]-to-lipid ratio after incubation at pH 4.5, 6, or 7.4 and the [ZnL(AcO)]-to-lipid ratio before incubation at pH 7.4, according to the following equation:(3)$$\mathrm{ZnL}\left(\mathrm{AcO}\right)\;\mathrm{Associated}\;\mathrm{to}\;\mathrm{Liposomes}\;\left(\%\right)=\frac{\left(\frac{\mathrm{ZnL}\left(\mathrm{AcO}\right)}{\mathrm{Lip}}\right)f,\;\mathrm{pH}=4.5/6/7.4}{\left(\frac{\mathrm{ZnL}\left(\mathrm{AcO}\right)}{\mathrm{Lip}}\right)i,\;\mathrm{pH}=7.4}\times 100$$

##### Internalization Cell Studies

HCT-116 and CT-26 spheroids were incubated 72 h after their formation with complete medium (Control) or Rh-labelled [ZnL(AcO)] liposomes at the lipid concentrations of 2 and 4 µmol/mL. The internalization of Rh-labelled [ZnL(AcO)] liposomes was evaluated 48 and 120 h after incubation. Briefly, the culture medium was removed, and spheroids were washed with PBS to remove non-associated liposomal formulations. Then, spheroids were transferred into 8-well optical chamber slides (Lab-Tek Chamber Slide System, Fisher Scientific) with 100 μL of PBS and analyzed immediately on a Zeiss LSM 710 confocal microscope with a 10× objective and 575–595 nm (Ex-Em) of rhodamine with a 561 nm laser. Z-stack images were obtained at fixed intervals of 12 μm from the periphery into the center of the spheroid.

##### In Vitro Safety Assay by Hemolytic Activity

The hemolytic activity was measured as previously described by Gaspar et al. [40]. Briefly, ethylene diamine tetraacetic acid (EDTA)-preserved peripheral human blood was collected from a voluntary donor and used on the same day for the experiments. The hemolytic activity of the zinc complex [ZnL(AcO)] was determined in the free and liposomal forms. The peripheral blood was centrifuged at 1000× *g* for 10 min to separate and remove the plasma from the erythrocytes. Subsequently, the erythrocytes were diluted in PBS and centrifuged at 1000× *g* for 10 min. This procedure was repeated three times. The metal-based complex in the free and liposomal forms was diluted in PBS to concentrations ranging from 5.5 to 700 µM and distributed in 96-well plates (100 μL/well). Moreover, 100 μL of erythrocyte suspension was added to all samples, incubated at 37 °C for 1 h, and centrifuged at 800× *g* for 10 min. The absorbance of the supernatants was measured at 550 nm with a reference filter at 620 nm using a BioTekTM ELx800TM absorbance microplate reader (Winooski, VT, USA). The percentage of the hemolytic activity for each sample was calculated by comparing with a positive control, corresponding to 100% hemolysis (erythrocytes in distilled water), and a negative control (erythrocytes in PBS) according to the following equation:(4)$$\mathrm{Hemolytic}\;\mathrm{Activity}\;\left(\%\right)=\frac{{\mathrm{Sample}}_{\mathrm{Abs}}-\mathrm{Negative}\;{\mathrm{Control}}_{\mathrm{Abs}}}{{\mathrm{Positive}\;\mathrm{Control}}_{\mathrm{Abs}}-{\mathrm{Negative}\mathrm{Control}}_{\mathrm{Abs}}}\times 100$$

#### 2.2.10. In Vivo Assays

##### Animals

Male 8–10-week-old C57Bl/6 mice were purchased from Charles River (Barcelona, Spain). The animals were kept under standard hygiene conditions, fed commercial chow, and given acidified drinking water ad libitum. All experiments were conducted according to the Animal Welfare Organ, ORBEA, of the Faculty of Pharmacy, Universidade de Lisboa, and approved by the competent national authority Direção-Geral de Alimentação e Veterinária (DGAV) in accordance with the EU Directive (2010/63/EU) and Portuguese laws (DR 113/2013, 2880/2015, and 260/2016).

##### Tumor Syngeneic Mouse Model

For tumor induction, a total of 1 × 10^6^ CT-26 murine colon cancer cells were suspended in PBS (100 μL) and injected subcutaneously (s.c.) in the right flank of male Balb/c mice [35]. Tumors became palpable around 13 days after tumor induction, and the treatment schedule was initiated. Mice were randomly divided in five groups (6 animals per group) and received daily intravenous (i.v.) injections of the formulations under study. Two negative control groups were established: mice receiving PBS (Control) or empty liposomes (Empty Lip). Moreover, the other three groups received i.v. injections of [ZnL(AcO)] in the free (Free [ZnL(AcO)]) or liposomal forms (F4) and 5-FU at 5 and 15 mg/kg body weight, respectively. The mice were monitored every day for signs of pain or distress. Body weight and tumor size were regularly monitored, and the latter was measured using a digital caliper. The respective tumor volumes were calculated according to the formula: V (mm^3^) = (L × W2)/2, where L and W represent the longest and shortest axes of the tumor, respectively. The relative tumor volumes (RTV) were determined for each animal as the ratio between the volumes at the indicated day and the volumes at the beginning of treatment. Two days after the last treatment, the animals were euthanized, and blood was collected. The primary tumors, spleen, liver, kidneys, and lungs were excised and weighed. The tissue index was calculated according to the following equation:(5)$$\mathrm{Tissue}\;\mathrm{Index}=\sqrt{\frac{\mathrm{organ}\;\mathrm{weight}}{\mathrm{animal}\;\mathrm{weight}}\times 100}$$

##### Hepatic Biochemical Parameters

Serum was isolated from the blood samples, and serum aspartate transaminase (AST) and serum alanine transaminase (ALT) were measured using a commercially available kit (Girona, Spinreact, Spain) according to the manufacturer’s specifications.

##### Statistical Analysis

GraphPad Prism Version 5.03 (GraphPad Software, San Diego, CA, USA) was used to perform all the statistical analyses. Statistical differences were evaluated with ANOVA, and the differences were considered to be significant for *p* < 0.05.

## 3. Results and Discussion

One new Schiff base derived from 8-hydroxyquinoline and benzothiazole as well as two new Zn(II) complexes with different L:M stoichiometries were obtained with good yields (>77%) and purity. All compounds were characterized in the solid state and solution with different spectroscopic and analytical techniques that allowed establishing their structural formulations.

### 3.1. The Ligand Precursor

The ligand precursor corresponds to a hydrazone resulting from the condensation of 2-carbaldehyde-8-hydroxyquinoline with 2-hydrazinobenzothiazole, as shown in Figure 1.

The product’s elemental analysis and ESI mass spectrum are in good agreement with the expected structural formula. The ^1^H NMR spectrum shows two singlets corresponding to the hydrazinic NH at δ_H_ = 12.75 ppm and the hydroxyl OH proton at δ_H_ = 9.82 ppm. At 8.34 ppm, a large peak integrating for two protons comprises one proton of the quinoline ring and the imine proton (Appendix A). The aromatic ring protons are found between 8.35 and 7.13 ppm. In the FTIR spectrum of HL (Appendix A), the sharp, strong band at 3411 cm^−1^ corresponds to the hydroxyl group. The imine C=N band appears at 1621 cm^−1^, while in the range of 1593–1566 cm^−1^ the C–N bands from the benzothiazole and quinoline moieties appear. The phenol C–O is present at 1144 cm^−1^, and the C–S from the benzothiazole is present at 745 cm^−1^. In the UV-Vis absorption spectrum of HL (Appendix A) in DMSO, it is possible to identify three bands with maxima at λ = 265 nm (ε = 2.60 × 10^4^ M^−1^cm^−1^), 368 nm (ε = 2.87 × 10^4^ M^−1^cm^−1^), and 476 nm (ε = 9.78 × 10^3^ M^−1^cm^−1^). The two higher energy bands correspond to π→π* transitions within the aromatic rings, while the band at 476 nm is associated with the hydrazone moiety.

### 3.2. The Zn(II) Complexes 

From reactions performed with different L:M ratios, two Zn(II) complexes were isolated, [ZnL(AcO)] and [ZnL_2_], according to an elemental analysis and mass spectrometry data. In the FTIR spectrum of [ZnL(AcO)], the band that was observed at 3411 cm^−1^ in the ligand precursor’s spectrum disappeared, indicating the deprotonation of the phenolic group and the coordination of the O-phenolate to the metal center (Appendix A). ^1^H NMR spectroscopy supports this binding mode since the signal appearing at 9.9 ppm in the complex spectrum has a carbon cross peak in the HSQC correlation spectrum at 144.9 ppm, confirming that it is not due to the hydroxyl but due to the imine proton (Appendix A). This peak suffered a downfield shift resulting from removal of electronic density by coordination to the zinc ion. Further support is provided by the FTIR bands associated with these group vibrations that decreased in intensity and are shifted to lower wavenumbers (11–12 cm^−1^) in the complex spectrum. Additionally, the presence of the methyl group from the coordinated acetate ion is observed in the NMR (–CH_3_ group in DMSO_*d*_6_, δ_H_ = 1.87 ppm, δ_C_ = 21.32 ppm). The combination of these observations with the elemental analysis and mass spectrometry results indicates an octahedral geometry for the [ZnL(AcO)] complex, with the ligand precursor acting as a mono-anionic tetradentate ligand through the deprotonated oxygen of the phenolic group and the nitrogen atoms from the hydroxyquinoline, imine, and benzothiazole moieties; the coordination sphere is completed by an acetate anion. In the UV-Vis absorption spectrum recorded in DMSO (Appendix A), the complex shows a red shift of the intraligand transition bands, accompanied by a decrease in their intensity (smaller ε values), confirming the coordination of the ligand to the metal center. For the 2:1 stoichiometry, the compound’s elemental analysis and mass spectrometry are in good agreement with the formula [ZnL_2_]. The NMR spectrum (Appendix A) shows the presence of the OH signal at the same chemical shift as in the ligand precursor’s spectra, implying that in this complex the coordination does not involve the phenolate moiety. This is corroborated by FTIR that shows the presence of the sharp OH band at 3411 cm^−1^. In addition, the NMR spectra do not present signals from the acetate ions, corroborating that each organic molecule is coordinated as mono-anionic. Since the hydroxyl group is protonated, the isomerization of the benzothiazole double bond has most likely taken place, and the coordination to zinc is accomplished by the benzothiazole and imine N-atoms. This can clarify the presence of the very strong sharp band at 1561 cm^−1^, which is due to the formed –C=N-N=C–. Considering all experimental evidence, the proposed structures for the Zn(II) complexes are the ones shown in Figure 2. 

### 3.3. Interaction with Bovine Serum Albumin

Investigations in recent years have revealed the importance of carrying out interaction studies of compounds developed for therapy with serum albumins for the analyses of their pharmacological profiles. Bovine serum albumin (BSA) has long been used as a model for its human homologue due to their structural similarity. Albumins are drug carriers that can help solve solubility and stability issues [41,42,43,44]. Spectrophotometric techniques were used to evaluate the effect of the presence of albumin on the solubility of the complexes and to evaluate the binding process. The studies are detailed in the Appendix A, and here we present only the main observations. Both zinc complexes present low solubility in aqueous media, precipitating heavily in concentrations as low as 15–30 µM in 0.6% DMSO/PBS (Appendix A). In the presence of equimolar amounts of BSA, only very small changes were observed for 24 consecutive hours by UV-Vis absorption (Appendix A), in contrast to the precipitation observed in its absence. Fluorescence spectroscopy was used to quantify the binding strength (see Appendix A). The obtained Stern–Volmer quenching constants were 1.7 × 10^5^ M^−1^ for [ZnL(AcO)] and 2.6 × 10^5^ M^−1^ for [ZnL_2_]. Since the Trp emission in albumins has a lifetime (τ_0_) in the nanoseconds range, this means that the calculated k_q_ values are above the collisional limit and, therefore, the quenching process has a static component from the formation of a complex between the Zn compounds and albumin. The application of the Scatchard equation to both systems yielded an association constant of 5.0 × 10^4^ M^−1^ for [Zn(L)(AcO], while [ZnL_2_] showed a constant more than an order of magnitude higher, *K* = 6.4 × 10^5^ M^−1^. In both cases, one binding site was obtained, with *n* = 0.91 and 1.04 for [ZnL(AcO)] and [ZnL_2_], respectively. These results are in good agreement with previously published data for 8-hydroxyquinoline derivatives [45] and other metal complexes bearing hydrazone ligands [46]. Association constants in the range of 10^4^–10^5^ M^−1^ are considered to represent moderate to strong binders, which indicates that the complexes can be transported in the serum by the protein [47] and released at the target site. Moreover, circular dichroism (CD) titrations of BSA (24 µM) with increasing amounts of each complex (Appendix A) revealed the appearance of induced CD signals, providing an additional solid proof of the interaction between the Zn complexes and BSA. 

### 3.4. Interaction Studies with DNA

Titrations of *ct*DNA with the complexes were followed by CD spectroscopy. However, low or no affinity of either complex to *ct*DNA were observed. Moreover, displacement assays with the classical intercalator ethidium bromide, carried out by fluorescence spectroscopy, showed the inefficiency of both complexes to compete for the same positions within the DNA (Appendix A). 

### 3.5. In Vitro Screening of the Zinc Complexes: 2D and 3D Settings

In order to assess the antiproliferative properties of the zinc complexes developed in this work, a MTT assay was performed with [ZnL(AcO)] and [ZnL_2_] against a non-tumorigenic cell line, the immortalized keratinocyte cell line HaCaT, and against the human and murine colon cancer cell lines, HCT-116 and CT-26, after 48 h of incubation in a monolayer culture. The results depicted in Table 1 indicate that, as expected, both molecules are potent antiproliferative agents. Moreover, both complexes present weak selectivity towards human cancer cell lines in comparison with the non-tumorigenic cell line. [ZnL(AcO)] appeared to achieve the most consistent IC_50_ values among the two colon cancer cell lines under study.

The use of cancer cells cultured in a 3D setting is a model that better resembles the behavior of a tumor in vivo and thus is more adequate to select potential anti-tumor agents [48]. In fact, Karlsson and collaborators have shown that HCT-116 spheroids generally present morphological, immunohistochemical, genetic, and pathophysiological properties that resemble solid tumors in patients, in opposition to monolayer cultures [49]. Thus, the LDH assay was used to determine the cytotoxicity of HCT-116 and CT-26 cells cultured in a 3D setting. CRC cell spheroids were incubated with [ZnL(AcO)] and [ZnL_2_] at 5, 10, and 20 µM for 72 h based on the IC_50_ values obtained in the 2D cell model (below 20 µM). The results are depicted in Figure 1.

Interestingly, the obtained data revealed that for both compounds tested at the highest concentration (20 µM) cytotoxicity only reached values of 20% in HCT-116 and CT-26 spheroids, which contrasts with the data obtained through the MTT assay performed in a 2D cell model. These results demonstrated that the same cells may behave in a completely different manner depending on their structural organization. This outcome is consistent with the literature, as it is thought that a 3D cellular organization allows cancer cells to obtain in vivo tumor-like features, such as drug resistance, that are influenced by gene expression modulation, cell–cell connections, and microenvironment alterations [48,49]. Many studies highlight spheroids as a much more accurate representation of cancer biology concerning cell heterogeneity, different drug levels, and oxygen diffusion [50,51,52]. The obtained data (Figure 1A) revealed that [ZnL(AcO)] can significantly increase the cytotoxicity when compared to the controls in HCT-116 and CT-26 spheroids. 

Furthermore, in both cell lines, [ZnL(AcO)] also led to a significant increase in the spheroids’ cytotoxicity when compared to [ZnL_2_], corroborating monolayer studies. Moreover, the obtained spheroids’ images (Figure 1B) also revealed solubility problems when higher concentrations of the zinc complexes were used, particularly in the case of [ZnL_2_]. This is in accordance with the literature concerning metal-based complexes and water solubility problems [4]. Consequently, [ZnL(AcO)] was selected for further experiments that are discussed below.

### 3.6. Cell Death and Cell Cycle Analysis

To better understand the mechanism behind the toxicity of the Zn(II) complex [ZnL(AcO)], two assays were performed to assess the type of cell-induced death and the compound’s impact on the cell cycle. HCT-116 and CT-26 cell lines, cultured in a 2D setting, were incubated with [ZnL(AcO)] at 20 µM for 24 h and further stained with P.I. and Annexin V for flow cytometry analysis. This concentration was based on the IC_50_ values obtained in Section 3.5 (2D cell model). Annexin V detects cellular apoptosis, while P.I. detects cells that already lost membrane integrity and are therefore dead. When cells were stained with both dyes, they were considered to be undergoing necrosis. Indeed, the obtained data (Figure 2) revealed that the type of cell death more commonly induced by the zinc complex was necrosis, particularly for CT-26 cell line. Moreover, after incubation with [ZnL(AcO)], only approximately 10 and 3% of the cell populations of the HCT-116 and CT-16 cells, respectively, remained viable. Thus, this experiment further supports the cytotoxic potential of this complex.

For the cell cycle analysis, the metal complex was tested at 5 and 10 µM (below IC_50_ values), aiming to achieve low or moderate effects on cell proliferation. Concerning the impact of the compound on the cell cycle (Appendix A), the data revealed that both HCT-116 and CT-26 cells incubated with [ZnL(AcO)] at 5 and 10 µM did not exhibit statistically significant differences in terms of cell population, in percentage, in the analyzed cell cycle phases in comparison to control cells incubated with complete medium. In addition, the proliferation indexes of both cell lines after incubation with the metal-based complex were determined based on the sum of the percentage of cells in the S and G2/M phases (Appendix A). Once again, no differences in proliferation index values were found between the cells incubated with [ZnL(AcO)] and the respective control. Thus, the results demonstrate that [ZnL(AcO)] may act through a cell-cycle-independent mechanism. Previous results from our group with a copper complex also revealed antiproliferative activity through a cell-cycle-independent mechanism [35].

Based on the spectroscopic and cell-based studies, the low solubility and low stability of the complexes in the micromolar range in biological media will probably have a strong impact on the therapeutic efficacy of metal-based complexes. This is a common problem of labile transition-metal ions, which can be circumvented following their association to drug delivery systems. This nanotechnological strategy may protect loaded metal-based complexes from premature degradation, improve their stability in biological media, solve solubility issues, and allow preferential targeting to affected sites [35].

### 3.7. Physicochemical Characterization of Liposomal Formulations

As a result of the promising cytotoxic profile of [ZnL(AcO)] towards tumor cells, its incorporation in liposomes was envisioned to passively deliver the compound to CRC sites, taking advantage of the EPR effect, as already mentioned. Different liposomal formulations were developed to achieve an optimal liposomal formulation that could efficiently incorporate the complex (Table 2). The influence of phospholipids with different phase transition temperatures (Tc) on the loading capacity of [ZnL(AcO)] in liposomes was evaluated. Specifically, two neutral phospholipids, DMPC and DOPC (synthetic phospholipids), were used as the major constituents of the liposomal formulations. These phospholipids present Tc values of +24 and −17 °C, respectively. Moreover, CHEMS and DOPE in combination and DSPE-PEG were included in the lipid composition to confer pH-sensitive properties and long blood-circulation times, respectively [35]. F1 presented low loading capacity and incorporation efficiency values (Table 2). This outcome may be a consequence of the highly hydrophobic nature of [ZnL(AcO)], which accumulates to a lower extent in the DMPC lipid bilayer with a Tc of 24 °C, an effect described in literature [53]. In F2, DMPC was replaced by DOPC, and the same lipid ratio and compound concentration were maintained. As DOPC has a much lower T_c_ than DMPC, thus being more fluid, it could facilitate the compound’s incorporation in the lipid bilayer.

Nevertheless, F2 also presented low loading capacity and I.E. values. In F3, a reduction in the molar ratio of CHEMS was tested, and the changed lipid ratio resulted in a higher amount of fluid phospholipids in the liposomes, which should once again facilitate the compound’s incorporation. The results demonstrate that F3 maintained a low loading capacity but slightly increased the incorporation efficiency. In an attempt to further enhance the loading capacity, another formulation was tested (F4), maintaining the same phospholipid concentration and lipid ratios as in F3 but increasing the initial compound concentration from 500 to 1000 µg/mL. This change in the experimental conditions resulted in a higher loading capacity as well as a high I.E., with 23 µg/µmol and 76% compared to F3 with 7 µg/µmol and 63%, respectively. Considering the size, it has been described in the literature that liposomes up to 200 nm can take advantage of the leaky tumor vasculature to reach cancer cells [54,55]. The liposomal formulations developed in this work ranged from 110 to 150 nm in size and presented high homogeneity, hence offering appropriate mean sizes to take advantage of the endothelial fenestrations that are characteristic of tumor sites. Regarding the zeta potential, all four formulations presented zeta potential values close to neutrality, ranging from −7 to −3 mV in accordance with the lipid composition used [34,35]. Considering the obtained data, the formulation selected for further studies was F4. 

### 3.8. Assessment of pH-Sensitive Properties of the Liposomal Formulation

To validate the pH-sensitive properties of the F4 nanoformulation and its ability to release the drug at the target site, the quantification of the zinc complex associated to nanoliposomes was assessed upon exposure to different pH conditions. Nanoformulations were incubated in HEPES buffer at pH 4.5, 6.0, or 7.4 to mimic the pH conditions of the tumor microenvironment and consequently test if once F4 reaches tumor sites [ZnL(AcO)] is released. According to the results depicted in Figure 3, an increase in [ZnL(AcO)] release from the nanoliposomes upon incubation at lower pH values was achieved. Moreover, the literature describes that CHEMS protonation occurs from pH values from 4.5 to 6.5, which is also in accordance with the obtained results [56]. However, lower percentages of [ZnL(AcO)] associated with liposomes at the lowest tested pH values (4.5) were expected. In fact, previous results from our group with a copper metal-based complex exhibited lower levels of compound associated with nanoliposomes at pH 4.5 [34]. In fact, at this pH, only ~20% of the copper complex remained associated to the liposomes [34], which contrasts with the 71% observed for [ZnL(AcO)]. This difference might be related to the fact that [ZnL(AcO)] is highly hydrophobic and consequently might have a stronger interaction with the lipid bilayer, hampering the pH-dependent release. In addition, the use of longer incubation period could result in a higher release. Moreover, the formulations used in the current work also possess a lower molar ratio of CHEMS, which may result in decreased pH sensitivity properties. 

### 3.9. Antiproliferative Properties of Liposomal Formulation

Following the optimization and selection of experimental conditions for the preparation of [ZnL(AcO)] liposomes, it was necessary to assess if the nanoformulation maintained the cytotoxic properties of the loaded compound. Hence, the viability of HCT-116 and CT-26 cells cultured in a 2D setting was evaluated in the absence (negative control) and presence of different concentrations of [ZnL(AcO)] in its free and liposomal forms using the MTT assay after 48 h of incubation. Additionally, 5-FU, a commonly used chemotherapeutic agent against CRC, was used as a positive control.

The results (Table 3) demonstrated that the developed liposomal formulation maintained the antiproliferative properties of the free metal complex against both CRC cell lines, i.e., the [ZnL(AcO)] activity was not affected after incorporation. Still, the use of long-blood-circulating pH-sensitive nanoliposomes for in vivo delivery is crucial for improving the antitumor effect of [ZnL(AcO)]. As already mentioned, the biodistribution profile of drugs following their association with long-blood-circulating liposomes can change and preferentially accumulate loaded compounds to a higher extent at tumor sites [57,58,59,60]. Moreover, in terms of the required doses to achieve a therapeutic effect, both the liposomal and free complex presented very low IC_50_ values when compared to 5-FU in the human colon cancer cell line, indicating that both forms of the zinc complex presented a higher cytotoxic potential than the commonly used chemotherapeutic agent [61].

### 3.10. Liposome Internalization Studies in Spheroids

Internalization studies of Rh-labelled liposomes incorporating [ZnL(AcO)] into HCT-116 and CT-26 spheroids were performed to assess the liposomes’ ability to penetrate colon cancer cells cultured in a 3D setting. Rh-labelled liposomes were prepared using the method described above by including in the lipid composition the rhodamine covalently linked to phosphatidyl ethanolamine. Since the zinc complex is quantified by spectrophotometry at a maximum absorbance wavelength of 393 nm, the addition of rhodamine did not allow the quantification of the metal compound, as it interfered with its absorbance, and only the lipid content was quantified. Hence, 72 h after formation, HCT-116 and CT-26 spheroids were incubated with Rh-labelled liposomes at the lipid concentrations of 2 and 4 µmol/mL for 48 and 120 h. Concerning the first time point of 48 h of incubation with both the human and murine colon cancer cell line spheroids, no liposome internalization was observed independent of the tested lipid concentration (data not shown). The 3D structural organization of cancer cells influences liposome internalization that needs more time to occur. Thus, liposome internalization into CRC spheroids was also assessed after 120 h of incubation (Figure 4). The data demonstrated that liposome internalization is time- and concentration-dependent since, at this time point, it was possible to observe that the outer layers of spheroid cells were rhodamine-positive, particularly when a higher concentration of liposomes was used (4 µmol/mL). Moreover, spheroids incubated with the higher lipid concentration were much smaller than the controls as well as the ones incubated with lower lipid concentrations. This occurred independently of the cell line used. Additionally, some of these spheroids had surrounding detached round cells that were also rhodamine-positive. This suggests that liposomes loaded with the metal-based complex are internalized from the outer layers of the spheroids towards the inner layers, most likely killing layer after layer of cells and consequently decreasing the spheroid size. Based on these results, the antiproliferative properties of [ZnL(AcO)] in a 3D setting, presented in next section, were observed following an incubation period of 120 h.

### 3.11. Antiproliferative Properties of [ZnL(AcO)] in a 3D Setting

To further analyze the antiproliferative properties of both the free and liposomal forms of [ZnL(AcO)], the cell viability of HCT-116 and CT-26 spheroids was assessed through the CellTiter-Glo^®^ 3D kit after 120 h of incubation (Figure 5A). In the first set of in vitro studies in a 3D cell model (Figure 1A), [ZnL(AcO)] in the free form exhibited very modest antiproliferative properties using a range of concentrations from 5 to 20 µM following an incubation period of 72 h. Based on liposome internalization studies (Section 3.10), it was demonstrated that longer incubation periods were needed. In this sense, the antiproliferative properties of both the liposomal and free forms of [ZnL(AcO)] were tested in a 3D setting at much higher concentrations and for 120 h. While in the case of [ZnL(AcO)] in the free form, solubility problems precluded the use of concentrations higher than 50 µM, in the liposomal form it was possible to test up to 200 µM. In fact, microscopy images (Figure 5B) showed that the free [ZnL(AcO)] complex tends to crystalize on top of the spheroids of both cell lines for the higher concentrations used. The lack of compound solubility and consequent precipitation may be considered a drawback, as it greatly impairs parenteral drug administration [4,62,63]. On the contrary, liposomal [ZnL(AcO)] did not present any crystallization at the tested concentrations ranging from 50 to 200 µM. This result demonstrates that liposomes are in fact able to solve drug solubility issues.

Furthermore, this experiment also demonstrated that in a 3D setting higher concentrations of [ZnL(AcO)] are needed to reduce cell viability, once again indicating that spheroids present higher drug resistance than cells cultured in monolayers. The incubation of [ZnL(AcO)] liposomes at 100 and 200 µM with HCT-116 and CT-26 spheroids resulted in a viability decrease of around 40 to 50%. These results contrast with in vitro data obtained for cells cultured in monolayers. Indeed, in a 2D setting, 10 µM [ZnL(AcO)] incorporated in liposomes was enough to decrease the cell viability by about 50%.

Karlsson and collaborators also evaluated the cytotoxic properties of irinotecan, 5-FU, and oxaliplatin towards HCT-116 in 3D and 2D model settings and observed a much higher resistance in the 3D model [49]. Thus, the choice of using spheroids in the present work reinforces the importance of using these models to study drug effects, as they tend to better represent the challenges of drug development encountered in in vivo experiments [48,49]. In addition, it is important to note that liposomes in vivo allow the use of higher drug concentrations, thus increasing the amount of drug at tumor sites, a strategy commonly adopted by oncologists. The liposomal formulation of doxorubicin, Doxil^®^, is a good example [64]. Indeed, this liposomal formulation is able to reduce the cardiotoxicity associated with this compound, thus allowing its therapeutic use at higher doses [59,64].

### 3.12. In Vitro Safety Assay of [ZnL(AcO)] 

The US FDA recommends that for all excipients intended for injectable use, an in vitro hemolysis study should be performed at the intended concentrations for intravenous (i.v.) administration [65]. Therefore, the hemolytic activity against RBCs was used as a marker of the general membrane toxicity effect of [ZnL(AcO)] in the free and liposomal forms. In this assay, the percentage of hemolysis of the complex in both forms was tested for concentrations up to 700 μM. As shown in Figure 6, hemolysis was always below 1.5% for all tested concentrations. Thus, this preliminary safety assay demonstrated that both the free and liposomal forms of the compound can be intravenously injected [66].

### 3.13. In Vivo Efficacy of [ZnL(AcO)] in a Syngeneic Colon Cancer Mouse Model

Considering the promising obtained in vitro results, the therapeutic potential of the chosen [ZnL(AcO)] formulation was subsequently evaluated in a syngeneic murine colon cancer model following the subcutaneous (s.c.) injection of the murine CT-26 colon cancer cell line in the right flank of Balb/c mice. (Figure 7).

As soon as the tumor mass became palpable in all animals, the treatment protocol was initiated. In detail, five experimental groups were established: Control (mice receiving PBS), Empty Lip (empty liposomes), 5-FU (5-Fluorouracil), [ZnL(AcO)] Free, and [ZnL(AcO)] Lip (F4 incorporating [ZnL(AcO)]). Treated mice received intravenous (i.v.) administrations of the tested formulations at a dose of 5 mg/kg of body weight, with the exception of 5-FU, which was administered at 15 mg/kg.

During the treatment schedule, several parameters were evaluated: relative tumor volume (RTV), animal weight, mouse health, and behavior. Through a RTV analysis that evaluated the tumor progression of each animal over time, a higher therapeutic effect was observed for animals receiving 5-FU, indicating impairment of tumor progression, compared to both control groups and to animals treated with free [ZnL(AcO)]. However, a similar antitumor effect was observed for animals that received the [ZnL(AcO)] liposomal formulation, even though a 3-fold lower dosage was used (Figure 7B). Concerning animal weight, a slight reduction throughout all experimental protocols was observed for all groups under study, particularly for the Control and Free [ZnL(AcO)] groups (Figure 7A). At the end of the treatment, the tumor mass weights were also determined (Figure 7C), thus corroborating the results from the RTV analysis. Indeed, animals treated with 5-FU and [ZnL(AcO)] Lip exhibited the lowest average tumor mass (<0.5 g), whereas 0.7, 1.1, and 1.0 g were achieved for the Control, Empty Lip, and Free [ZnL(AcO)] groups, respectively. The average tumor mass for the group treated with free [ZnL(AcO)] was 2.4-fold higher in comparison to animals that received the metal-based complex in the liposomal form, thus reinforcing the importance of the association of the metal complex with a lipid-based nanosystem. Finally, the safety of the [ZnL(AcO)] formulations at the tested dose of 5 mg/kg was evaluated through the tissue index and hepatic biomarker determination (Figure 7D). In terms of the tissue index, no significant changes were observed for the analyzed organs among the studied groups. Moreover, all values were within the reference intervals [35]. The tissue index allows the estimation of organ development, and increased values can be a consequence of organ hypertrophy, congestion, or edema, while decreased tissue indexes indicate organ atrophy and degenerative changes [67,68]. Concerning the hepatic aminotransferases alanine aminotransferase (ALT) and aspartate aminotransferase (AST), alterations in the normal values provide information that allows the detection or prediction of potential toxic hepatic effects after drug exposure [69]. The obtained results were also within the reference intervals for the animal strain and in accordance with previous studies [35]. Overall, these results attest to formulation safety for i.v. administration.

## 4. Conclusions

In this study, a new Schiff base incorporating two bioactive moieties, 8-hydroxyquinoline and benzothiazole, and two Zn(II) complexes were successfully developed and characterized. Both zinc-based complexes displayed promising IC_50_ values against human and murine colon cancer lines when cultured in monolayers. Once cancer cells were cultured in a 3D cell model, much higher concentrations were required to impact the cell viability. Still, [ZnL(AcO)] displayed a higher antiproliferative capacity than [ZnL_2_] and was incorporated in long-blood-circulating and pH-sensitive liposomes with an appropriate size to take advantage of the EPR effect. In this last case, formulation F4 impaired tumor progression, as revealed by the RTV values in comparison to mice that received the compound in the free form. Moreover, the average tumor mass at the end of the experimental protocol for the mouse group treated with the metal-based nanoformulation was similar to animals treated with 5-FU using a 3-fold lower dosage. In addition, the tissue index and hepatic biomarker determination support the safety of the developed nanoformulation. These results obtained in this study constitute a proof of concept. Further in vivo studies should be carried out, namely, pharmacokinetic, biodistribution, and safety studies. In addition, other colon cancer murine models that most closely mimic what occurs in humans should also be tested, as well as different treatment doses and schedules. Overall, our results show the pivotal importance of designing new metal complex scaffolds and controlling their low solubility as well as bioavailability through their incorporation in appropriate drug delivery systems. The therapeutic strategy proposed here paves the way for the development of more effective and safer treatment approaches for CRC treatment.

## Data Availability

The datasets generated for this study are available on request to the corresponding authors.

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
