# Peer review of "Liposomal Formulations of a New Zinc(II) Complex Exhibiting High Therapeutic Potential in a Murine Colon Cancer Model"

_ijms, 2022, doi:10.3390/ijms23126728_

Round 1

Reviewer 1 Report

The manuscript entitled "Liposomal formulations of a new zinc(II) complex exhibiting high therapeutic potential in a murine colon cancer model" describes the antiproliferative potencial of zinc-based complexes. Complexes were screened in vitro using murine and human colon cancer 2D and 3D cells culture. Moreover, these complexes were evaluated for anticancer activity in vivo in tumor syngeneic mouse model. Overall, the manuscript is technically sound and well written but there are some questions that need to be clarified.

      The abstract is lengthy. I suggest it should be clear and concise. There should be harmony in reference style.

2     It is not clear how authors selected the concentrations here tested in cytotoxicity (bibliography, preliminary work?). Furthermore, there is no clear indication of the relationship between concentrations analysed in cytotoxicity (MTT assay) using 2D cell culture (5 to 40 µM) and 3D cell culture (5 to 20 µM), ATP quantification (20 to 200 µM), apoptosis (20 µM) and cell cycle analysis (5 to 10 µM). Why does duration of exposure changes from one assay to other (24 to 120 h)? The methodological plan is somehow difficult to understand as choices are not suitably justified.

 3      Also on this matter, there is some confusing information regarding HaCaT cell line. HaCaT cell line was used to evaluate cytotoxicity of metal complexes by MTT assay (section 3.5) but subsequent tests were carried out only on CT-26 and HCT-116 cell lines. Is there an explanation for this? Moreover, there is no information about HaCaT cell line in experimental section.

Author Response

The manuscript entitled "Liposomal formulations of a new zinc(II) complex exhibiting high therapeutic potential in a murine colon cancer model" describes the antiproliferative potential of zinc-based complexes. Complexes were screened in vitro using murine and human colon cancer 2D and 3D cells culture. Moreover, these complexes were evaluated for anticancer activity in vivo in tumor syngeneic mouse model. Overall, the manuscript is technically sound and well written but there are some questions that need to be clarified.

  1. The abstract is lengthy. I suggest it should be clear and concise. There should be harmony in reference style.

Reply: We have followed the reviewer suggestion and the abstract was reduced and changed aiming to be clearer and more concise.

  1. It is not clear how authors selected the concentrations here tested in cytotoxicity (bibliography, preliminary work?). Furthermore, there is no clear indication of the relationship between concentrations analysed in cytotoxicity (MTT assay) using 2D cell culture (5 to 40 µM) and 3D cell culture (5 to 20 µM), ATP quantification (20 to 200µM), apoptosis (20 µM) and cell cycle analysis (5 to 10 µM). Why does duration of exposure changes from one assay to other (24 to 120 h)? The methodological plan is somehow difficult to understand as choices are not suitably justified.

Reply:

  1. The in vitro studies performed in this manuscript have been done step by step in an interactive way; that is why the concentrations and incubation periods tested for the two metal-based complexes were selected based on the results that have been achieved in the different in vitro
  2. In section 3.5 the first set of in vitro assays were performed: first in 2D models and then based on the IC50 values in a 3D setting. For the MTT assay the concentrations tested ranged from 5 to 40 µM and based on the IC50 values: below 20 µM; the in vitro assays in a 3D model were then performed using the maximum concentration of 20 µM. The conjugation of the 2D and 3D cell models allowed to select the metal-based complex [ZnL(AcO)] for further studies.
  3. In the second set of in vitro experiments, section 3.6, the elucidation of the mechanisms of action of the selected metal-based complex was performed. For cell death studies the concentration tested was 20 µM corresponding to a concentration slightly higher than the IC50 values observed for the [ZnL(AcO)] (section 3.5) in a 2D cell model. For the cell cycle analysis, the concentrations tested (5 and 10 µM) were based on the fact that, in this type of assay, low or moderate effects on cell proliferation should occur and so selected concentrations must be below the IC50
  4. D) In section 3.10 – liposome internalization studies in spheroids, the data obtained led us to the conclusion that longer incubation times should be used, to allow liposomes to penetrate colon cancer cells cultured in 3D settings. The obtained results demonstrated that at least 120 h were necessary, this being the reason for the long incubation period selected for antiproliferative studies in 3D cell model.
  5. E) Moreover, in the first set of in vitro studies in a 3D cell model (Figure 1A) [ZnL(AcO)] in the free form exhibited very modest antiproliferative properties using a range of concentration from 5 to 20 µM. Based on these results higher concentrations were tested. While in the case of [ZnL(AcO)] in the free form, solubility problems restricted the use of concentrations higher than 50 µM, in the liposomal form it was possible to test this metal-based complex up to 200 µM.

We have followed all reviewer comments and in each in section, the justification of the experimental conditions namely concentrations and incubation periods were included in the revised version of the manuscript.

  1. Alsoon this matter, there is some confusing information regarding HaCaT cell line. HaCaT cell line was used to evaluate cytotoxicity of metal complexes by MTT assay (section 3.5) but subsequent tests were carried out only on CT-26 and HCT-116 cell lines. Is there an explanation for this? Moreover, there is no information about HaCaT cell line in experimental section.

Reply: The present work aims to test and validate the antitumor potential of the 2 metal-based complexes. In the first set of in vitro assays our objective was to evaluate the antiproliferative properties and selectivity of the zinc-based complexes towards colon cancer cell lines. For this reason, HaCat was included in the in vitro assays, as a non-tumorigenic cell line for evaluating the selectivity of the zinc-based complexes.

The following in vitro studies aimed to elucidate the mechanisms of action of the two metal-based complexes as well as the in vitro behavior in 3D models. This was the reason why the following assays were only performed in the murine and human colon cancer cell lines.

In the experimental section information about HaCat cell line was included.

Reviewer 2 Report

The submitted manuscript entitled - Liposomal formulations of a new zinc(II) complex exhibiting high therapeutic potential in a murine colon cancer model - focuses on the development of a liposomal format of a newly synthesized hydrophobic anticancer drug and the test platform evaluating the therapeutic relevance. The manuscript is well structured, easy to follow,  the used methodology is adequate, and the results are well discussed with the current literature. Supplementary files are well done.

Remaining questions

(1)   Zn(II) complexes-BSA association was determined as moderate to strong binders. Might this binding behavior hamper cell toxicity studies

(2)   Liposomes as drug carriers are known to increase the availability of drugs, especially hydrophobic drugs. It´s obviously known that the lipid composition, but also the lamellarity, significantly influence the encapsulation efficiency. The selected preparation method results in unilamellar structures with limited entrapment capacities. Would other nanocarriers maybe more appropriate for the transport of the Zn complexes, especially because the assumed pH release is also limited.

(3)   Is there any evidence how long these liposomes are stable during storage because of the Zn-complex located in the membrane

(4)   Is the Tm of the membranes significantly modulated by the Zn-complex incorporation in respect to the hydrophobicity and concentration

(5)   free [ZnL(AcO)] complex tends to crystalize on top of the spheroids of both cell lines-why does this occur although they should be attached to serum proteins

(6)   In vivo efficacy of liposomal [ZnL(AcO)] has been demonstrated. Is the resulted anti-tumor effect sufficient for a potent anticancer candidate.

(7)   Is this formulation really safer in respect to repeated and maybe high dose treatment regimes

Reviewer comments should be reflected in the text of the manuscript.

Author Response

Suggestions for Authors

The submitted manuscript entitled - Liposomal formulations of a new zinc(II) complex exhibiting high therapeutic potential in a murine colon cancer model - focuses on the development of a liposomal format of a newly synthesized hydrophobic anticancer drug and the test platform evaluating the therapeutic relevance. The manuscript is well structured, easy to follow,  the used methodology is adequate, and the results are well discussed with the current literature. Supplementary files are well done.

Remaining questions

  1. Zn(II) complexes-BSA association was determined as moderate to strong binders. Might this binding behavior hamper cell toxicity studies

Reply: The binding of a prospective drug to albumin is a well-studied parameter that aims to evaluate its transport in plasma and its distribution to target sites. It also allows to increase the solubility of hydrophobic substances in the plasma. Other authors have also observed that the presence of BSA in mammal incubation cell media may impact the antiproliferative assays outcome, since it may help solubilize the compounds, but very strong binding may hamper the release of the compound from the proteins. For moderate to strong binders, as the ones included in the current work, this should not be a problem. Moreover, in a previous published paper, we have addressed this issue (https://doi.org/10.1016/j.jinorgbio.2019.110727). Indeed we observed that binding to albumin in the culture media decreased the toxicity of the Zn-complexes, which was consistent with previous data from Haase et al. (https://doi.org/10.1039/c4mt00206g).

Liposomes as drug carriers are known to increase the availability of drugs, especially hydrophobic drugs. It´s obviously known that the lipid composition, but also the lamellarity, significantly influence the encapsulation efficiency. The selected preparation method results in unilamellar structures with limited entrapment capacities. Would other nanocarriers maybe more appropriate for the transport of the Zn complexes, especially because the assumed pH release is also limited.

Reply: The in vitro release of loaded zinc-based complex at pH 4.5 was around 30% after 90 min of incubation. We can speculate that prolonged incubation periods could demonstrate a higher drug release. This assumption is based on the in vitro assays performed in 3D cell model that clearly demonstrated that prolonged incubation periods were more effective in terms of cytotoxicity. Unfortunately, such prolonged release studies were not performed. Despite this, of course there may be other alternatives for nanocarriers for this type of compounds and we are currently testing others namely those of polymeric nature.

  1. Is there any evidence how long these liposomes are stable during storage because of the Zn-complex located in the membrane

Reply: We agree with the reviewer about the importance of the stability of zinc-based complex in liposomes. However, liposomes are very stable formulations (Gaspar et al, 2015-doi.org/10.1016/j.nano.2015.06.008; Pinho et al, 2021- https://doi.org/10.1016/j.ijpharm.2021.120463). These issues should constitute further studies to be performed in near future.

  1. Is the Tm of the membranes significantly modulated by the Zn-complex incorporation in respect to the hydrophobicity and concentration

Reply: We completely agree that calorimetric studies are very important assays to evaluate if the presence of zinc-based complex within the lipid bilayer results in alterations in terms of thermotropic behavior of liposomes. Moreover, this may give insights of the stability of zinc-based complex liposomal formulation. The main aim of this study was to assess the efficacy of new synthetized compounds and their delivery through a liposomal formulation, but we will take this subject into consideration in the next studies.

  1. free [ZnL(AcO)] complex tends to crystalize on top of the spheroids of both cell lines-why does this occur although they should be attached to serum proteins

Reply: The binding of the complexes to serum albumin is a reversible process, meaning that there is free and bound complex molecules in equilibrium. Moreover, cells were incubated in complete medium that contains 10% FBS, corresponding to ca. 40 mM of BSA (https://doi.org/10.1016/j.jinorgbio.2019.110727). In the 3D antiproliferative assays, the loss of solubility and the appearance of crystals occurred for concentrations superior to 50 mM, which is higher than the average amount of BSA in a medium with 10% FBS. Thus, above 40 mM the molecules precipitate since they do not have enough serum protein to bind.

 In vivo efficacy of liposomal [ZnL(AcO)] has been demonstrated. Is the resulted anti-tumor effect sufficient for a potent anticancer candidate.

Reply: The results included in the present manuscript constitute a proof of concept of the selected metal-based complex following its incorporation in liposomes. Although the high antitumor effect was observed several additional studies should be performed, namely, pharmacokinetic biodistribution studies and safety in vivo assays. In the present work a subcutaneous colon cancer murine model was used. The antitumor activity should also be assessed in other colon cancer murine models that mimic what occurs in humans. We really appreciate the positive feedback of the reviewer.

  1. Is this formulation really safer in respect to repeated and maybe high dose treatment regimes

Reply: In the present work a single therapeutic dose was tested. However, other doses and treatment schedules should be performed in a very near future to answer this high relevant question.

  1. Reviewer comments should be reflected in the text of the manuscript.

Reply: Authors are very grateful for the positive feedback. Throughout the manuscript several sentences were included to account for the issues raised by the reviewer. Moreover, in the conclusions section and based on the comments of the review we have included the following sentences:

“The very good results obtained in this study constitute a proof of concept. Further in vivo studies should be carried out namely pharmacokinetic, biodistribution and safety studies. In addition, other colon cancer murine models that mimic better what occurs in humans should also be tested as well as different treatment doses and schedules”.

Round 2

Reviewer 1 Report

The authors addressed adequately all my comments. I have no additional comments.